# Carbon Emissions from Manufacturing Sector in Jiangsu Province: Regional Differences and Decomposition of Driving Factors

**Ping Zhou and Hailing Li ***

Business School, Hunan Agricultural University, Changsha 410128, China; zhou_ping2022@163.com
* Correspondence: hilynn_lee1027@csu.edu.cn

**Abstract:** Based on the Tapio decoupling model, this paper discusses the decoupling relationship between the economic growth and carbon emissions of the manufacturing sector in southern Jiangsu, northern Jiangsu and middle Jiangsu during the 13th Five-Year-Plan period. By using the LMDI method, the carbon emissions and influencing factors of 31 subindustries of the manufacturing sector in Jiangsu Province from 2016 to 2020 were quantitatively analyzed by region and industry. The main findings are as follows: (1) during the 13th Five-Year-Plan period, the growth rate of the energy consumption and carbon emissions of the manufacturing sectors in southern Jiangsu, northern Jiangsu and middle Jiangsu slowed down, and the industrial structure was increasingly optimized; (2) economic growth is the primary driving force behind the manufacturing carbon emissions in the three regions of Jiangsu Province, while energy intensity is the main factor that affects the carbon-emission differences among the manufacturing subsectors in the different regions; (3) improving the energy efficiency of high-emission-intensity industries, such as the ferrous metal smelting and calendering industry, chemical industry and textile industry, is the key to reducing the carbon emissions of the manufacturing sector in the different regions of Jiangsu in the future. Jiangsu Province should promote the upgrading of the manufacturing-industry structure, and it should encourage the high-energy-consumption industry to reduce its energy intensity by technological innovation to achieve the goal of emission reduction and economic growth.

**Keywords:** Tapio decoupling; carbon emissions; low-carbon transformation; sustainable development; Jiangsu



## 1. Introduction

In September 2020, President Xi Jinping of the People's Republic of China pointed out, at the 75th United Nations General Assembly, that China's carbon dioxide emissions will peak by 2030, and that China will achieve carbon neutrality by 2060. To achieve this goal, the Chinese government proposed, in the 14th Five-Year Plan, to support those provinces with conditions that can take the lead to reach the carbon-emission peak, requiring them to formulate action plans for carbon-emission peaking before 2030 [1,2].

Jiangsu, which is located in the Yangtze River Economic Belt, has 13 prefecture-level cities, and it is the only province where all the prefecture-level cities are among the top 100 in China (see Figure 1). In 2021, the added value of manufacturing in Jiangsu Province accounted for 35.8% of the regional gross domestic product (GDP) and 13.3% of the national GDP, ranking it first in China [3,4]. Furthermore, as the largest manufacturing province in China, Jiangsu Province has 40 industrial sectors, and its extensive development is accompanied by large amounts of energy consumption and carbon emissions. During the 13th Five-Year-Plan period, the industrial energy consumption accounted for more than 70% of the total energy consumption in Jiangsu, and $SO_2$ and $NO_x$ accounted for 86% and 52% of the province, respectively [4]. The implementation of the national strategic deployment of "carbon peak and carbon neutrality", and the realization of low-carbon and

green economic development, are important tasks that are faced by Jiangsu Province in the "14th Five-Year Plan" period.

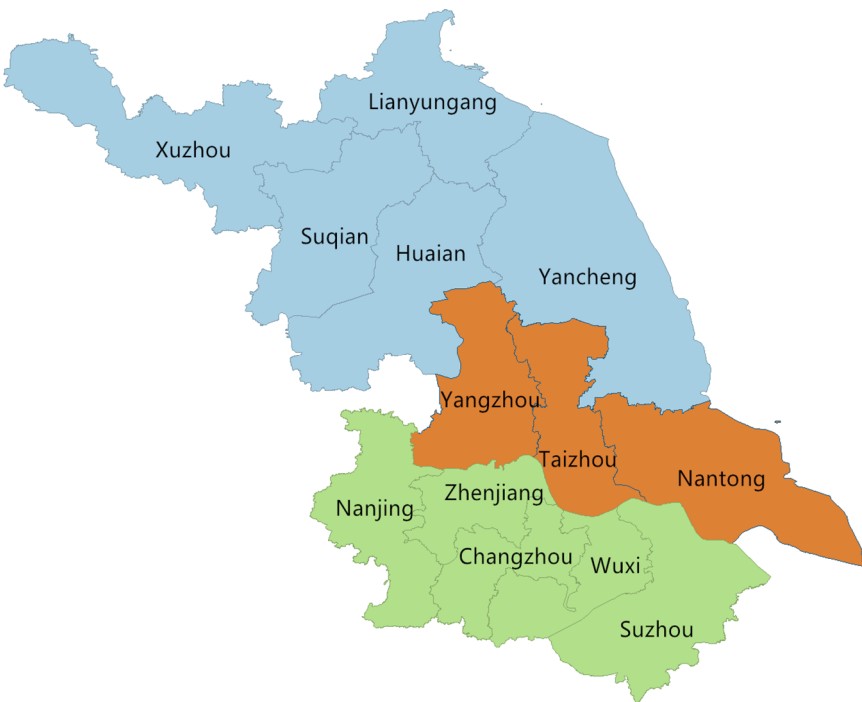

**Figure 1.** Map of Jiangsu Province (northern Jiangsu in blue, middle Jiangsu in yellow and southern Jiangsu in green).

The existing studies have been paying attention to carbon emissions for a long time, and there have been extensive discussions on the exponential decomposition of the influencing factors of carbon-emission change. However, most studies are conducted at the national level, and there are few studies that specifically focus on the driving factors of carbon emissions in provincial and regional manufacturing industries. Based on different geographical locations and levels of economic development, Jiangsu has formed three regions: southern Jiangsu, northern Jiangsu and middle Jiangsu (seen in Figure 1). There are large differences in the levels of green regional development in Jiangsu Province, with the highest green-development index in southern Jiangsu, the second highest in middle Jiangsu and the lowest in northern Jiangsu. Therefore, researching the decoupling relationship and the main driving factors of the manufacturing carbon emissions and industrial development in different regions is of great significance to the realization of low-carbon transformation and development in Jiangsu Province.

Based on existing research, this paper analyzes the decoupling statuses of the carbon emissions in the manufacturing subindustries of southern Jiangsu, northern Jiangsu and middle Jiangsu from 2016 to 2020, based on the Tapio decoupling model. We adopted the LMDI decomposition method, which measures the main driving factors of carbon emissions in the manufacturing subindustries of different regions. The main contributions of this paper are as follows: (1) the decomposition results are extended to the provincial regional level and industry level, and the decoupling index is used to explore the regional and industry differences in the carbon emissions in Jiangsu Province from a more detailed dimension; (2) a subindustry LMDI decomposition model was constructed to reveal the main driving forces of the carbon-emission differences among the three regional manufacturing subindustries, so as to provide a reference for the study of carbon-emission-reduction schemes for the manufacturing sector in Jiangsu Province. This has important reference significance for Jiangsu Province and will allow it to formulate economic adjustment and energy policies under different industrial sectors, clarify the responsibility and obligation

of carbon-emission reduction, transform the economic growth mode and promote the harmonious development of the economy and environment.

　　The rest of this paper is as follows: In Section 2, we present the literature review; we present the methodology in Section 3; in Section 4, we present the results and discussions; in Section 5, we present the conclusion and policy suggestions.

## 2. Literature Review

　　The decoupling between carbon emissions and economic growth is considered to be the key to achieving green economic development. The linkage between economic growth and environmental pollution or resource consumption was first described by the Organization for Economic Cooperation and Development (OECD) as "decoupling" [5]. Traditionally, researchers have explored the main factors that affect energy consumption and environmental pollution based on different methods, including the frameworks of index decomposition analysis (IDA) [6,7] and structural decomposition analysis (SDA) [8,9]. Derived from the framework of IDA, the LMDI model has been widely applied to evaluate the efficiency in the energy and environmental sectors [10–12]. In addition, based on the study of Tapio [13], eight possible decoupling statuses were analyzed, and researchers have investigated and analyzed the decoupling statuses at various research levels with the Tapio model combined with LMDI theory [14]. In recent years, decoupling theory has been widely used for research on the decoupling relationship between economic growth and carbon emissions.

　　Many existing studies have focused on the country level [15–24] and region or city levels [25–29], exploring different decoupling statuses and the factors that affect economic growth and carbon emissions. At the country level, Chen et al. [30] indicated that the main factors that affect the carbon dioxide emissions of OECD countries were the energy intensity and per capita GDP, based on the LMDI model and the Kaya identity framework. Shuai et al. [31] found that higher-income-level groups had more possibilities of reaching their desired decoupling statuses than lower-income-level groups after identifying the income-level groups and decoupling statuses of 133 countries. In contrast to the above studies, Wang and Su [22] explored the impacts of urbanization and industrialization on decoupling by the Granger causality test and Johansen cointegration theory. Song et al. [32] used the case of China and the United States to distinguish the decoupling statuses of regions at different economic-development levels by studying the internal economic relationship between the decoupling model and the environmental Kuznets curve hypothesis. At the regional or city level, Dong et al. [33] decomposed the decoupling index into eight affecting factors at the regional level, with the investigation of the spatial and temporal heterogeneities in the influencing factors in each province of China. Wang et al. [34] comparatively studied the decoupling statuses, trends and effects of carbon emissions from economic growth between Beijing and Shanghai from the whole and sectoral perspectives. Liu et al. [35] analyzed the decoupling relationship between economic growth and industrial carbon emissions from 13 prefecture-level cities in Jiangsu Province.

　　Some researchers have studied the decoupling of output growth from carbon emissions at the sector level, such as in the agricultural industry [36], aircraft-related industry [37], metal industry [38,39], construction industry [40], manufacturing industry [41], power industry [42,43], transportation industry [34,40,44–46] and the whole industry [47–50]. Lu et al. [51] divided the industry into three main departments that consist of 38 subindustries, based on the importance of the sectoral dimension, and they analyzed the main factors that affect the energy-related industrial carbon emissions in Jiangsu, which is one province of China. Yang et al. [52] analyzed the decoupling elasticity and factors of industrial growth and carbon emissions in different Chinese regions, and they evaluated the contributions of different sectors. Wen and Li [53] analyzed the drivers of the industrial carbon emissions in 30 provinces of China based on optimized spectral clustering and proposed CE-reduction strategies. In addition, some scholars have also analyzed the decoupling statuses and trends between the industrial added value and carbon emissions from the perspectives of different

industrial subindustries. For example, Song et al. [54] studied the decoupling status and CE-reduction potential of China's transport sector, and they pointed out that the main impact factor was the economic-growth effect. Hang et al. [55] explored the decoupling status and affecting factors between the carbon emissions and industrial added values of manufacturing considering the heterogeneity in China's manufacturing subindustries.

These existing studies have analyzed and evaluated the decoupling statuses and trends between economic variables and carbon emissions from the country, regional and city levels, and from a department perspective. However, few studies have comparatively researched the changes in the decoupling statuses and trends in manufacturing from different regions. Given this, based on the Tapio decoupling framework and the LMDI model, this study explores the decoupling of carbon emissions and manufacturing industrial added value and its affecting factors from a regional perspective. Furthermore, this paper evaluates the decoupling statuses, trends and affecting factors of the key subindustries by considering the heterogeneities in the regions and their policies. This research provides the theoretical basis for the low-carbon development of Jiangsu's different regions and different manufacturing subindustries, and it provides a policy reference for the overall coordinated development of manufacturing in Jiangsu, and even the whole country.

## 3. Methodology

### 3.1. LMDI Model

This paper selects the extended LMDI as the decomposition model of carbon emissions from the industrial sector $i$, which can be expressed as Equation (1):

$$
\begin{aligned}
C_i &= \sum_j C_{ij} = \sum_j \left( \frac{C_{ij}}{E_{ij}} \times \frac{E_{ij}}{IN_{ij}} \times \frac{E_i}{IN_i} \times \frac{IN_i}{G} \times G \right) \\
&= \sum_j \left( CE_{ij} \times ES_{ij} \times EI_{ij} \times IG_i \times G \right)
\end{aligned}
\tag{1}
$$

where $C_i$ represents the carbon emissions from the industrial sector $i$; $C_{ij}$ represents the carbon emissions caused by the fossil energy consumption of energy $j$ from the industrial sector $i$; $E_{ij}$ represents the fossil energy consumption of energy $j$ from the industrial sector $i$; $E_i$ represents the energy consumption of the industrial sector $i$; $IN_i$ represents the industrial added value of the industrial sector $i$; $G$ represents the gross regional GDP; $CE_{ij}$ represents the CE factor of energy $j$ from the industrial sector $i$; $ES_{ij}$ represents the ratio of the energy $j$ consumption to total energy consumption; $EI_{ij}$ represents the energy intensity of the industrial sector $i$; $IG_i$ represents the ratio of industrial added value of the industrial sector $i$ to the region's GDP.

The change in carbon emissions in a period $t$, $\Delta C_i$ is decomposed into five affecting factors, as shown in Equation (2) (carbon emissions, energy structure, energy intensity, industrial structure and industrial development), by using the LMDI model:

$$
\Delta C_i = \Delta C_{i,CE} + \Delta C_{i,ES} + \Delta C_{i,EI} + \Delta C_{i,IG} + \Delta C_{i,G}
\tag{2}
$$

The five driving factors can also be expressed by Equations (3)–(8):

$$
\Delta C_{i,CE} = \sum_j L(C_{i,j}^t, C_{i,j}^{t-1}) \ln(CE_{i,j}^t / CE_{i,j}^{t-1})
\tag{3}
$$

$$
\Delta C_{i,ES} = \sum_j L(C_{i,j}^t, C_{i,j}^{t-1}) \ln(ES_{i,j}^t / ES_{i,j}^{t-1})
\tag{4}
$$

$$
\Delta C_{i,EI} = \sum_j L(C_{i,j}^t, C_{i,j}^{t-1}) \ln(EI_{i,j}^t / EI_{i,j}^{t-1})
\tag{5}
$$

$$
\Delta C_{i,IG} = \sum_j L(C_{i,j}^t, C_{i,j}^{t-1}) \ln(IG_{i,j}^t / IG_{i,j}^{t-1})
\tag{6}
$$

$$
\Delta C_{i,G} = \sum_j L(C_{i,j}^t, C_{i,j}^{t-1}) \ln(G_{i,j}^t / G_{i,j}^{t-1})
\tag{7}
$$

$$L(C_{i,j}^t, C_{i,j}^{t-1}) = \begin{cases} (C_{i,j}^t - C_{i,j}^{t-1})/(lnC_{i,j}^t - \ln C_{i,j}^{t-1}), C_{i,j}^t \neq C_{i,j}^{t-1} \\ C_{i,j}^t, C_{i,j}^t = C_{i,j}^{t-1} \end{cases} \tag{8}$$

### 3.2. Tapio Decoupling Model

Based on the decoupling coefficient defined by Tapio (2005) [13], the decoupling elasticity of the industry *i* can be measured as Equation (9):

$$D_i = \frac{\Delta C_i/C_i^0}{\Delta GDP_i/GDP_i^0} \tag{9}$$

By combining the LMDI model with the decoupling method, the decoupling status and coefficient between carbon emissions and industrial added value in different manufacturing sectors in different regions of Jiangsu are analyzed. By combining Equation (2) with Equation (9), the elasticity $D_i$ is decomposed into five factors:

$$\begin{aligned} D_i &= \frac{(\Delta C_{i,CE} + \Delta C_{i,ES} + \Delta C_{i,EI} + \Delta C_{i,IG} + \Delta C_{i,G})/C_i^0}{\Delta GDP_i/GDP_i^0} \\ &= \frac{\Delta C_{i,CE}/C_i^0}{\Delta GDP_i/GDP_i^0} + \frac{\Delta C_{i,ES}/C_i^0}{\Delta GDP_i/GDP_i^0} + \frac{\Delta C_{i,EI}/C_i^0}{\Delta GDP_i/GDP_i^0} + \frac{\Delta C_{i,IG}/C_i^0}{\Delta GDP_i/GDP_i^0} + \frac{\Delta C_{i,G}/C_i^0}{\Delta GDP_i/GDP_i^0} \\ &= d_{CE} + d_{CE} + d_{CE} + d_{CE} + d_{CE} \end{aligned} \tag{10}$$

where $d_{CE}$, $d_{ES}$, $d_{EI}$, $d_{IG}$ and $d_G$ represent the five determinants for the decoupling index of the industry (i): the elasticity of the CE-intensity factor, energy-structure (ES) factor, energy-intensity (EI) factor, industrial-structure (IG) factor and industrial-development (G) factor.

In this paper, all kinds of energy uses are converted into corresponding standard coal, and the CE coefficient (C) generated by the complete combustion of standard coal per ton is calculated according to the fixed value recommended by the Energy Research Institute of the National Development and Reform Commission. Therefore, only $d_{EI}$, $d_{IG}$, and $d_G$ are included in the decomposition factor for the decoupling index of industry *i*.

Based on the decoupling-elastic-coefficient (*D*) value, the decoupling status can be classified into three categories and eight subcategories, as shown in Table 1 (Tapio, 2005) [13].

**Table 1.** The Tapio decoupling statuses.

| Decoupling Status | | % C | % GDP | D |
|---|---|---|---|---|
| Negative decoupling | Expansive Negative Decoupling (END) | + | + | (1.2, +∞) |
| | Weak Negative Decoupling (WND) | − | − | [0, 0.8) |
| | Strong Negative Decoupling (SND) | + | − | (−∞, 0) |
| Decoupling | Recessive Decoupling (RD) | − | − | (1.2, +∞) |
| | Weak Decoupling (WD) | + | + | [0, 0.8) |
| | Strong Decoupling (SD) | − | + | (−∞, 0) |
| Coupling | Expansive Coupling (EC) | + | + | [0.8, 1.2] |
| | Recessive Coupling (RC) | − | − | [0.8, 1.2] |

### 3.3. Data Source

This paper analyzes the decoupling between the industrial added value of manufacturing and carbon emissions in southern Jiangsu, northern Jiangsu and middle Jiangsu, based on the Tapio model. According to the industry classification of the Industrial Classification for National Economic Activities (ICNEA) (NBSC, 2017), this paper divides the manufacturing industry into 31 subindustries, and it investigates the heterogeneities in the decoupling statuses and trends of the manufacturing subindustries between the carbon emissions and industrial added value in different regions. In addition, according to the industrial-added-value proportion of the total manufacturing industrial added value, the

decoupling factors of the key subindustries are analyzed in different regions of Jiangsu. The data on economic change and industrial energy consumption were collected from the statistical yearbooks of JSY during the period from 2016 to 2020.

## 4. Results and Discussions

*4.1. Comparison of Decoupling Statuses, Trends and Decomposition Factors of Manufacturing in Three Regions*

### 4.1.1. Decoupling Statuses and Trends of Manufacturing in Different Regions

According to the results of Table 2, during the 13th Five-Year-Plan period, the decoupling statuses of the manufacturing in southern Jiangsu, middle Jiangsu and northern Jiangsu were different [7,30,34,40,56]. Moreover, the decoupling status of middle Jiangsu was similar to that of northern Jiangsu. Specifically, the manufacturing in southern Jiangsu exhibited expansive negative decoupling in 2016–2017, weak decoupling in 2017–2018, strong decoupling in 2018–2019 and expansive negative decoupling in 2019–2020. The decoupling status of manufacturing in northern Jiangsu in 2016–2017 exhibited strong decoupling, expansive coupling in 2018–2019, and expansive negative decoupling in 2019–2020; the decoupling elastic coefficients of northern Jiangsu were $-0.4875$ in 2017–2018 and 3.4339 in 2019–2020. The reason may be the implementation of differentiated regional development policies. To promote and attain coordinated regional development, the Jiangsu provincial government put forward clear policies for supporting manufacturing development in northern Jiangsu and middle Jiangsu.

**Table 2.** The decoupling statuses of the manufacturing industries in southern Jiangsu, northern Jiangsu and middle Jiangsu.

|  |  | % C | % GDP | D | Status |
|---|---|---|---|---|---|
| Southern Jiangsu | 2016–2017 | 0.1198 | 0.0529 | 2.2665 | END |
|  | 2017–2018 | 0.0002 | 0.0795 | 0.0028 | WD |
|  | 2018–2019 | −0.0573 | 0.0251 | −2.2842 | SD |
|  | 2019–2020 | 0.0902 | 0.0021 | 42.4151 | END |
| Northern Jiangsu | 2016–2017 | −0.1186 | 0.3398 | −0.349 | SD |
|  | 2017–2018 | −0.0974 | 0.1997 | −0.4876 | SD |
|  | 2018–2019 | 0.1124 | 0.1357 | 0.8279 | EC |
|  | 2019–2020 | 0.4184 | 0.1218 | 3.4339 | END |
| Middle Jiangsu | 2016–2017 | −0.0768 | 0.1332 | −0.5766 | SD |
|  | 2017–2018 | −0.1264 | 0.1732 | −0.7297 | SD |
|  | 2018–2019 | 0.1034 | 0.0872 | 1.1853 | EC |
|  | 2019–2020 | 0.225 | 0.0982 | 2.2926 | END |

### 4.1.2. Comparison of the Decomposition Factors among the Three Regions

This paper further analyzes the factors that affect the decoupling statuses of the manufacturing in these three regions based on the LMDI model. As can be seen from Table 3, the energy intensity (d_EI) was the main driving factor that affected the decoupling elastic coefficients in southern Jiangsu, middle Jiangsu and northern Jiangsu, followed by the industrial-development factor (d_G). From a timespan perspective, the decoupling elastic coefficients in northern Jiangsu and middle Jiangsu showed an upward trend, while they showed a downward trend during 2016–2019 in southern Jiangsu. The energy-intensity factors in northern Jiangsu and middle Jiangsu changed from negative to positive, the industrial-development factors of the three regions were generally positive and the industrial-structure factors were quite different from each other. The proportion of industrial-development factors exceeded the proportion of energy-intensity factors in 2018-2019, which was when the statuses of northern Jiangsu and middle Jiangsu exhibited coupling.

**Table 3.** The decoupling factors affecting the decoupling elasticity in southern Jiangsu, northern Jiangsu and middle Jiangsu.

| | Period | d_EI | d_IG | d_G | D |
|---|---|---|---|---|---|
| Southern Jiangsu | 2016–2017 | 1.2348 | −0.01 | 1.0417 | 2.2665 |
| | 2017–2018 | −0.9595 | 0.0083 | 0.9541 | 0.0028 |
| | 2018–2019 | −3.2433 | 0.0062 | 0.9529 | −2.2842 |
| | 2019–2020 | 41.3717 | 0.1497 | 0.8936 | 42.4151 |
| Northern Jiangsu | 2016–2017 | −1.1577 | 0.069 | 0.7397 | −0.349 |
| | 2017–2018 | −1.3542 | 0.0163 | 0.8503 | −0.4876 |
| | 2018–2019 | −0.1615 | 0.0365 | 0.9529 | 0.8279 |
| | 2019–2020 | 2.3043 | 0.0378 | 1.0917 | 3.4339 |
| Middle Jiangsu | 2016–2017 | −1.4789 | −0.0353 | 0.9376 | −0.5766 |
| | 2017–2018 | −1.5924 | 0.0109 | 0.8518 | −0.7297 |
| | 2018–2019 | 0.1778 | 0.0114 | 0.9962 | 1.1853 |
| | 2019–2020 | 1.235 | 0.0158 | 1.0418 | 2.2926 |

According to the above empirical analysis, in northern Jiangsu and middle Jiangsu, which were more affected by policies, industrial-development factors and industrial-structure factors played greater roles, and the industrial development in southern Jiangsu was still more dependent on energy. Due to the policy requirements for environmental protection and industrial-structure upgrading, the manufacturing subindustries in the three regions still faced severe decoupling pressures.

*4.2. Analysis of the Decoupling Statuses, Trends and Decomposition Factors of Key Subindustries in the Three Regions*

This paper analyzed the decoupling statuses and trends of each manufacturing subindustry in three regions. Tables A1–A4 show the decoupling elastic coefficients of 31 manufacturing subindustries in southern Jiangsu, northern Jiangsu and middle Jiangsu from 2016 to 2020 (see the Appendix A, Tables A1–A4). In general, the decoupling statuses of the manufacturing development were not the same among the regions, but they tended to be the same during the research period, and simultaneously, the number of subindustries transitioning from the decoupling status to the negative decoupling status increased. This indicates that, although the green development of the manufacturing sector in Jiangsu Province achieved periodic achievements in the 13th Five-Year-Plan period, there was still a certain gap between the green-development level of the manufacturing industry and developed countries and advanced regions in China. For example, energy-intensive industries accounted for 80% of the industrial energy consumption, and coal consumption accounted for more than 50%.

Due to limited space, this paper selected the subindustries that accounted for more than 10% of the total manufacturing industrial added value as the key subindustries for the analysis of the decoupling statuses, trends, and decomposition factors. Therefore, three manufacturing subindustries were chosen in each region, including subindustries 26 (chemical raw materials and chemical products), 31 (nonmetallic mineral products) and 39 (communication equipment, computers and other electronic equipment) in southern Jiangsu; subindustries 26 (chemical raw materials and chemical products), 31 (nonmetallic mineral products) and 36 (equipment for special purposes) in northern Jiangsu; and subindustries 17 (textiles), 26 (chemical materials and chemical products) and 37 (transportation equipment) in middle Jiangsu.

4.2.1. Analysis of the Decoupling Statuses and Trends of Key Subindustries

Table 4 shows the decoupling statuses, trends and decomposition factors of these subindustries in different regions from 2011 to 2015.

**Table 4.** The decoupling statuses of key manufacturing subindustries in three regions.

| Region | Industry | Period | Proportion | % C | % GDP | D | Status |
|---|---|---|---|---|---|---|---|
| Southern Jiangsu | 26 | 2016–2017 | 13.64% | 0.3244 | −0.0130 | −24.8926 | SND |
| | | 2017–2018 | 9.59% | −0.0944 | −0.3329 | 0.2835 | WND |
| | | 2018–2019 | 9.89% | 0.0853 | 0.0470 | 1.8163 | END |
| | | 2019–2020 | 10.13% | 0.0692 | −0.0143 | −4.8292 | SND |
| | 31 | 2016–2017 | 16.71% | 0.4571 | −0.0760 | −6.0176 | SND |
| | | 2017–2018 | 15.77% | 0.0944 | −0.1044 | −0.9038 | SND |
| | | 2018–2019 | 14.93% | −0.0849 | −0.0392 | 2.1671 | RD |
| | | 2019–2020 | 13.15% | 0.1018 | −0.1522 | −0.6688 | SND |
| | 39 | 2016–2017 | 18.59% | −0.3489 | 0.2112 | −1.6518 | SD |
| | | 2017–2018 | 18.88% | −0.1132 | −0.0364 | 3.1106 | RD |
| | | 2018–2019 | 19.32% | −0.0128 | 0.0378 | −0.3392 | SD |
| | | 2019–2020 | 18.69% | 0.0786 | −0.0683 | −1.1510 | SND |
| Northern Jiangsu | 26 | 2016–2017 | 11.20% | 0.0774 | 0.0990 | 0.7824 | WD |
| | | 2017–2018 | 12.70% | −0.0617 | 0.2173 | −0.2838 | SD |
| | | 2018–2019 | 11.71% | 0.0054 | 0.0044 | 1.2215 | END |
| | | 2019–2020 | 11.45% | 0.7221 | 0.1515 | 4.7666 | END |
| | 31 | 2016–2017 | 12.97% | −0.4341 | 0.1160 | −3.7436 | SD |
| | | 2017–2018 | 11.11% | −0.0864 | −0.0803 | 1.0767 | RC |
| | | 2018–2019 | 10.12% | 0.3854 | −0.0080 | −48.1417 | SND |
| | | 2019–2020 | 10.05% | 0.5420 | 0.1698 | 3.1923 | END |
| | 36 | 2016–2017 | 10.64% | −0.1190 | 0.4759 | −0.2500 | SD |
| | | 2017–2018 | 11.53% | 0.0008 | 0.1638 | 0.0050 | WD |
| | | 2018–2019 | 10.59% | −0.0027 | 0.0002 | −16.1790 | SD |
| | | 2019–2020 | 17.27% | 0.5022 | 0.9216 | 0.5449 | WD |
| Middle Jiangsu | 17 | 2016–2017 | 10.67% | −0.1410 | 0.0015 | −96.2740 | SD |
| | | 2017–2018 | 11.01% | −0.0078 | 0.0222 | −0.3494 | SD |
| | | 2018–2019 | 9.99% | −0.0170 | −0.0762 | 0.2235 | WND |
| | | 2019–2020 | 9.39% | 0.0478 | 0.0597 | 0.8008 | EC |
| | 26 | 2016–2017 | 16.00% | 0.0450 | 0.1851 | 0.2432 | WD |
| | | 2017–2018 | 16.88% | −0.0826 | 0.0454 | −1.8185 | SD |
| | | 2018–2019 | 17.37% | 0.1235 | 0.0475 | 2.5987 | END |
| | | 2019–2020 | 14.30% | 0.0265 | −0.0713 | −0.3720 | SND |
| | 37 | 2016–2017 | 14.28% | 3.8703 | −0.0397 | −97.4965 | SND |
| | | 2017–2018 | 12.83% | −0.9376 | −0.1099 | 8.5334 | RD |
| | | 2018–2019 | 12.76% | 0.0799 | 0.0124 | 6.4478 | END |
| | | 2019–2020 | 12.18% | 1.0600 | 0.0770 | 13.7650 | END |

In southern Jiangsu, the carbon emissions of subindustries 26 and 31 completely increased, and those of subindustry 39 decreased. However, the industrial added values of all these subindustries decreased in total. In 2017–2018, subindustry 26 exhibited weak negative decoupling, subindustry 31 exhibited strong negative decoupling and subindustry 39 exhibited strong decoupling. In 2019–2020, subindustries 26, 31 and 39 all exhibited strong negative decoupling. During the research period, subindustry 26 always exhibited negative decoupling, subindustry 31 mainly exhibited negative decoupling and subindustry 39 mainly exhibited decoupling. The decoupling statuses of all three subindustries exhibited strong negative decoupling overall, and the industrial added values from all three subindustries fluctuated; however, carbon emissions rose in subindustries 26 and 31. In comparison, the decoupling status of subindustry 39 was better than those of the other two industries.

In northern Jiangsu, the carbon emissions and industrial added value of the three subindustries completely increased. In 2017–2018, subindustry 26 exhibited weak decoupling; subindustry 31 exhibited recessive decoupling, which was a large change from the

previous year of 2012; subindustry 36 exhibited weak decoupling. In 2019–2020, subindustry 26 exhibited expansive negative decoupling, which was similar to the previous year; subindustry 31 exhibited expansive negative decoupling; subindustry 36 exhibited weak decoupling. During the study period, subindustry 36 was always in a decoupling status, while subindustries 26 and 31 exhibited expansive negative decoupling overall. The carbon emissions and industrial added value of the three subindustries showed trends of fluctuating increases; in comparison, subindustry 36 achieved a better decoupling of carbon emissions and industrial added value.

In middle Jiangsu, the carbon emissions of subindustries 26 and 37 completely increased, and those of subindustry 17 decreased, although the industrial added value of all these subindustries increased overall. In 2017–2018, subindustry 17 exhibited strong decoupling, subindustry 31 exhibited weak decoupling and subindustry 37 exhibited strong negative decoupling. In 2019–2020, subindustry 17 exhibited expansive coupling, subindustry 31 exhibited expansive negative decoupling and subindustry 37 exhibited expansive negative decoupling. Subindustries 17 and 31 both changed greatly from the previous year. During the research period, the decoupling statuses of the three subindustries changed significantly.

In all, for the key subindustries of the manufacturing sector in southern Jiangsu, northern Jiangsu and middle Jiangsu, the decoupling statuses of the subindustries in southern Jiangsu had negative decoupling statuses for the longest time, which was quite different from the other two regions. Through the regional coordinated development policy of promoting the transfer of the factors of production between different regions, the decoupling statuses of the manufacturing subindustries in the three regions gradually converged, and the level of coordinated development among the regions continued to improve.

4.2.2. Analysis of the Decomposition Factors of Key Industries in Three Regions

First, this paper analyzed and compared the decomposition factors of subindustry 26 in the three regions. According to the results of Table 5, the proportion of the subindustry 26 industrial added value in the three regions fluctuated downward, and the variation trends in the decoupling elastic coefficients were quite different. Specifically, during the 13th Five-Year-Plan period, the energy intensity was the main factor that influenced subindustry 26 in southern Jiangsu, while the industrial structure and industrial development were the main factors that influenced subindustry 26 in northern Jiangsu. The factors of energy intensity and industrial structure had greater influences on the carbon emissions of subindustry 26 in middle Jiangsu.

**Table 5.** The decomposition factors of subindustry 26 (chemical raw materials and chemical products) in southern Jiangsu, northern Jiangsu and middle Jiangsu from 2016 to 2020.

| Region | Period | d_EI | d_IG | d_G | D | Status |
|---|---|---|---|---|---|---|
| Southern Jiangsu | 2016–2017 | −26.0549 | 1.166 | −0.0038 | −24.8926 | SND |
| | 2017–2018 | −0.8742 | 1.0073 | 0.1504 | 0.2835 | WND |
| | 2018–2019 | 0.7979 | 0.6962 | 0.3221 | 1.8163 | END |
| | 2019–2020 | −5.8709 | −1.7043 | 2.746 | −4.8292 | SND |
| Northern Jiangsu | 2016–2017 | −0.2077 | 0.1023 | 0.8877 | 0.7824 | WD |
| | 2017–2018 | −1.1605 | 0.5622 | 0.3145 | −0.2838 | SD |
| | 2018–2019 | 0.221 | −18.6649 | 19.6654 | 1.2215 | END |
| | 2019–2020 | 3.5296 | −0.1967 | 1.4337 | 4.7666 | END |
| Middle Jiangsu | 2016–2017 | −0.6948 | 0.7678 | 0.1702 | 0.2432 | WD |
| | 2017–2018 | −2.7554 | 1.1374 | −0.2004 | −1.8185 | SD |
| | 2018–2019 | 1.5625 | 0.6375 | 0.3986 | 2.5987 | END |
| | 2019–2020 | −1.4231 | 2.761 | −1.7099 | −0.372 | SND |

According to the above empirical results, as a major chemical province, the industrial level of the chemical industry in Jiangsu Province has yet to be optimized, and the

environmental-protection level of the enterprises is not high enough. This shows that the low-carbon transformation of the chemical industry in the 14th Five-Year-Plan period still has a long way to go, and it is urgent to systematically reconstruct a green-chemical-industry system that conforms to the laws of industrial development.

Second, this paper analyzed and compared the decomposition factors of subindustry 31 in southern Jiangsu and northern Jiangsu. Table 6 shows the analysis results. It can be seen from Table 6 that the proportion of the subindustry 31 industrial added value in these two regions showed a downward trend, while the elastic coefficient of its decoupling changed in the opposite direction. According to the results of Table 6, the energy intensity was also the main influencing factor on the carbon emissions in subindustry 31.

**Table 6.** The decomposition factors of subindustry 31 (nonmetallic mineral products) in southern Jiangsu and northern Jiangsu from 2016 to 2020.

| Region | Period | d_EI | d_IG | d_G | D | Status |
|--------|--------|------|------|-----|---|--------|
| Southern Jiangsu | 2016–2017 | −7.2804 | 1.2635 | −0.0007 | −6.0176 | SND |
| | 2017–2018 | −2.009 | 0.5782 | 0.5271 | −0.9038 | SND |
| | 2018–2019 | 1.191 | 1.331 | −0.3549 | 2.1671 | RD |
| | 2019–2020 | −1.808 | 0.8766 | 0.2626 | −0.6688 | SND |
| Northern Jiangsu | 2016–2017 | −4.465 | 0.1651 | 0.5563 | −3.7436 | SD |
| | 2017–2018 | 0.08 | 1.8368 | −0.8401 | 1.0767 | RC |
| | 2018–2019 | −49.3287 | 13.8796 | −12.6926 | −48.1417 | SND |
| | 2019–2020 | 2.0363 | −0.0491 | 1.205 | 3.1923 | END |

Furthermore, this paper decomposed the decoupling elastic coefficients of the remaining key subindustries in the three regions, and analyzed the main affecting factors, such as subindustry 39, subindustry 36, subindustry 17 and subindustry 37 (seen in Table 7). Specifically, energy intensity was the main factor that affected the decoupling elasticity of subindustry 39 in southern Jiangsu during the 13th Five-Year-Plan period, while industrial structure and industrial development were the main factors of subindustry 36 from 2018 to 2019 in northern Jiangsu. At other times, energy intensity was the most influential factor. In middle Jiangsu, the energy-intensity factor was still the main factor behind the decoupling elastic coefficients of subindustry 17 and subindustry 37.

During the 13th Five-Year-Plan period, southern Jiangsu accelerated the development of subindustry 39 (communication equipment, computers and other electronic-equipment industries), and the decoupling of the industrial added value and carbon emissions was achieved. Northern Jiangsu actively promoted the development of subindustry 36 (special-equipment manufacturing). During the research period, this subindustry mainly exhibited decoupling, and the decoupling of the carbon emissions and industrial added value was also achieved. The trend of decoupling in the superior subindustry in middle Jiangsu was not obvious, and high-quality development was not fully realized.

**Table 7.** The decomposition factors of other subindustries in southern Jiangsu, northern Jiangsu and middle Jiangsu from 2016 to 2020.

| Region | Industry | Period | d_EI | d_IG | d_G | D |
|---|---|---|---|---|---|---|
| Southern Jiangsu | 39 | 2016–2017 | −2.3895 | 0.7375 | 0.0002 | −1.6518 |
| | | 2017–2018 | 2.1507 | −0.4026 | 1.3624 | 3.1106 |
| | | 2018–2019 | −1.3144 | 0.5937 | 0.3815 | −0.3392 |
| | | 2019–2020 | −2.2270 | 0.4971 | 0.5789 | −1.1510 |
| Northern Jiangsu | 36 | 2016–2017 | −1.0183 | 0.6013 | 0.1670 | −0.2500 |
| | | 2017–2018 | −0.9215 | 0.4957 | 0.4308 | 0.0050 |
| | | 2018–2019 | −17.1776 | −512.1090 | 513.1075 | −16.179 |
| | | 2019–2020 | −0.3297 | 0.6557 | 0.2189 | 0.5449 |
| Middle Jiangsu | 17 | 2016–2017 | −97.2011 | −18.5937 | 19.5208 | −96.274 |
| | | 2017–2018 | −1.3346 | 1.4109 | −0.4257 | −0.3494 |
| | | 2018–2019 | −0.8078 | 1.2636 | −0.2323 | 0.2235 |
| | | 2019–2020 | −0.1935 | −1.0687 | 2.0630 | 0.8008 |
| Middle Jiangsu | 37 | 2016–2017 | −99.9910 | 4.3925 | −1.8980 | −97.4965 |
| | | 2017–2018 | 8.1753 | 0.3289 | 0.0292 | 8.5334 |
| | | 2018–2019 | 5.4148 | −0.4641 | 1.4971 | 6.4478 |
| | | 2019–2020 | 12.3520 | −0.8803 | 2.2932 | 13.7650 |

Notes: subindustry 39 represents communication equipment, computers and other electronic equipment; subindustry 36 represents equipment for special purposes; subindustry 17 represents textiles; subindustry 37 represents transportation equipment.

## 5. Conclusions and Policy Implications

### 5.1. Research Conclusions

This paper calculated the decoupling statuses between the manufacturing industry's carbon emissions and industrial added value in southern Jiangsu, northern Jiangsu and middle Jiangsu from 2016 to 2020, based on the Tapio model, and it explored the driving factors of carbon emissions that affect the decoupling status of each manufacturing subindustry with the LMDI model. The main findings are as follows:

1. During the 13th Five-Year-Plan period, the coordinated development level of the three regions gradually improved, but there was still a large gap. There were also significant differences in the carbon emissions and carbon-emission intensities among the different subsectors of the manufacturing industry in southern Jiangsu, northern Jiangsu and middle Jiangsu;

2. Industrial development is the most important driving factor of the manufacturing carbon emissions in southern Jiangsu, northern Jiangsu and middle Jiangsu, and especially for the industries with high-emission intensities, which are represented by ferrous metal smelting and calendering, the chemical industry and the textile industry; the contribution of the economic-activity effect to carbon emissions is the most significant;

3. Energy intensity is the most important driving force of carbon-emission reduction, and the most important influencing factor on the carbon-emission differences among the manufacturing subindustries in southern Jiangsu, northern Jiangsu and middle Jiangsu. The energy-intensity gap between high- and low-energy-intensity industries is further widening, and there is still a lot of room for improvement in the energy efficiencies of traditional high-emission-intensity industries.

### 5.2. Policy Recommendations

Based on these findings, the policy implications are as follows:

1. There is a need to clarify the carbon-peaking tasks of the key industries in different regions of Jiangsu Province, and to support key industries and enterprises to take the lead in achieving carbon peaking. The government should regard the ferrous-metal-smelting and calendering industry, with high total carbon emissions and high-

carbon-emission intensity, as the top priority of the carbon-peak work during the 14th Five-Year-Plan period. According to the characteristics of the industrial structures in different regions, the government should specify the carbon-peak tasks for the key industries in each region, and it should formulate relevant policy documents to guide enterprises to improve their energy efficiencies and reduce their carbon-emission intensities;

2. By combining the characteristics of the different regions in middle Jiangsu, southern Jiangsu and northern Jiangsu, the government should formulate differentiated carbon-peaking and carbon-neutrality action plans. Different regions should include the "3060" dual carbon target in their development plans, and they should actively explore the realization path to carbon peak and carbon neutrality. For example, the south of Jiangsu Province, with its developed economy and high industrial concentration, is a key area of energy consumption and carbon emissions, and it should be strongly encouraged to achieve green development through the adoption of new technologies and procarbon emissions;

3. There is a need to accelerate the development and application of energy-saving technologies and promote the optimization and upgrading of the industrial structure of Jiangsu Province. The upgrading of the industrial structure is an important means to promote energy conservation and emission reduction, but the industrial structure is unchangeable in the short term. Therefore, reducing carbon emissions by improving the technical level is the key to energy conservation and emission reduction in the Jiangsu manufacturing industry, through vigorously promoting scientific and technological innovation, promoting the industrial-structure upgrading of Jiangsu Province, promoting regional coordinated development and finally realizing the green economic transformation of Jiangsu Province.

*5.3. Limitations and Future Research*

Similar to most studies, some improvements could be made to this research. The energy-consumption data come from larger subindustries than small and medium enterprises, and so the overall values are smaller than the actual data. On the other hand, the LMDI model cannot reflect the changes in the factor structure and technological level for the whole economic field but can only reflect the changes in direct energy use. Therefore, due to the limitations of the method we have chosen, it is difficult to determine the influence of carbon emissions from one subindustry on those from other subindustries. However, our method fits our data, and the results are reliable. Further studies are still needed to determine the decoupling relationship between carbon emissions and the industrial added value from other inputs, and to calculate the direct and indirect impacts between different subindustries with other methods, such as SDA, which captures the greater impact of the economic structure and is easily extended to multiple regions.

In future research, some new methods should be tried to analyze the more specific driving factors of carbon emissions, such as the combination of DEA–LMDI based on input–output data; meanwhile, it is necessary to attempt to select more provinces and regions with different characteristics as research objects, such as Guangdong, the Yangtze River Delta region and the Pearl River Delta region, which could obtain more comprehensive research results to put forward more constructive suggestions.

**Author Contributions:** Conceptualization, H.L.; methodology, H.L.; formal analysis, P.Z. and H.L.; data curation, P.Z.; writing—original draft preparation, P.Z. and H.L.; writing—review and editing, P.Z. and H.L. All authors have read and agreed to the published version of the manuscript.

**Funding:** This research received no external funding.

**Institutional Review Board Statement:** Not applicable.

**Informed Consent Statement:** Not applicable.

**Data Availability Statement:** Not applicable.

**Conflicts of Interest:** The authors declare that they have no known competing financial interests or personal relationships that could have appeared to influence the work reported in this paper.

## Appendix A

**Table A1.** The decoupling statuses of manufacturing subindustries of three regions in 2016–2017.

| Period | Industry | Southern Jiangsu | | | | Northern Jiangsu | | | | Middle Jiangsu | | | |
|---|---|---|---|---|---|---|---|---|---|---|---|---|---|
| | | % C | % GDP | D | State | % C | % GDP | D | State | % C | % GDP | D | State |
| 2016–2017 | 13 | 0.0238 | 0.0881 | 0.2703 | WD | −0.0628 | −0.0273 | 2.3026 | RD | 0.1206 | −0.0091 | −13.2353 | SND |
| 2016–2017 | 14 | −0.0729 | −0.0093 | 7.8735 | RD | −0.1038 | 0.0495 | −2.0980 | SD | 0.0601 | 0.0789 | 0.7618 | WD |
| 2016–2017 | 15 | −0.9859 | −0.7222 | 1.3652 | RD | −0.0267 | 0.0463 | −0.5773 | SD | −0.0326 | −0.0610 | 0.5339 | WND |
| 2016–2017 | 16 | −0.0207 | 0.0000 | 0.0000 | SD | −0.0480 | 0.2918 | −0.1643 | SD | – | – | – | SD |
| 2016–2017 | 17 | −0.0371 | −0.2419 | 0.1535 | WND | −0.0848 | −0.0521 | 1.6274 | RD | −0.1410 | 0.0015 | −96.2740 | SD |
| 2016–2017 | 18 | 0.0548 | 0.2357 | 0.2325 | WD | −0.4263 | −0.1699 | 2.5099 | RD | 0.6224 | 1.9206 | 0.3241 | WD |
| 2016–2017 | 19 | 0.0465 | 0.1043 | 0.4457 | WD | −0.0600 | 0.0194 | −3.0928 | SD | −0.7793 | 0.2771 | −2.8127 | SD |
| 2016–2017 | 20 | −0.1331 | −0.0578 | 2.3007 | RD | −0.0572 | 0.0433 | −1.3221 | SD | −0.2735 | −0.1114 | 2.4561 | RD |
| 2016–2017 | 21 | −0.0918 | 4.1195 | −0.0223 | SD | 158.1224 | 18.0095 | 8.7800 | END | 0.0103 | −0.0056 | −1.8372 | SND |
| 2016–2017 | 22 | 0.0699 | 0.0815 | 0.8578 | EC | 0.0387 | −0.0599 | −0.6467 | SND | −0.0536 | −0.0017 | 31.1985 | RD |
| 2016–2017 | 23 | 0.1258 | −0.1713 | −0.7346 | SND | −0.0038 | 0.0772 | −0.0491 | SD | −0.9334 | −0.9806 | 0.9519 | RC |
| 2016–2017 | 24 | −0.2986 | 0.0534 | −5.5936 | SD | −0.4235 | −0.1924 | 2.2012 | RD | −0.9536 | 0.0934 | −10.2073 | SD |
| 2016–2017 | 25 | −0.0907 | 0.0100 | −9.0538 | SD | −0.1642 | −0.0011 | 149.2618 | RD | 0.0375 | 0.1084 | 0.3458 | WD |
| 2016–2017 | 26 | 0.3244 | −0.0130 | −24.8926 | SND | 0.0774 | 0.0990 | 0.7824 | WD | 0.0450 | 0.1851 | 0.2432 | WD |
| 2016–2017 | 27 | 0.0406 | 0.4305 | 0.0942 | WD | 0.5103 | 0.0666 | 7.6668 | END | −0.2045 | −0.0390 | 5.2495 | RD |
| 2016–2017 | 28 | −0.1046 | −0.0636 | 1.6446 | RD | −0.2601 | 0.0427 | −6.0902 | SD | −0.0586 | −0.0509 | 1.1521 | RC |
| 2016–2017 | 29 | −0.3309 | 0.1487 | −2.2261 | SD | 0.1030 | 0.2298 | 0.4482 | WD | 0.0137 | 0.0906 | 0.1507 | WD |
| 2016–2017 | 30 | 0.0073 | 0.2272 | 0.0322 | WD | −0.1478 | −0.0804 | 1.8375 | RD | −0.3165 | −0.0773 | 4.0956 | RD |
| 2016–2017 | 31 | 0.4571 | −0.0760 | −6.0176 | SND | −0.4341 | 0.1160 | −3.7436 | SD | −0.5571 | −0.0795 | 7.0065 | RD |
| 2016–2017 | 32 | −0.7078 | −0.0196 | 36.0664 | RD | 0.0483 | −0.2336 | −0.2069 | SND | −0.1483 | −0.0012 | 128.3436 | RD |
| 2016–2017 | 33 | −0.0767 | 0.0071 | −10.8125 | SD | −0.0646 | −0.2569 | 0.2517 | WND | 0.0497 | 0.0141 | 3.5247 | END |
| 2016–2017 | 34 | 0.1475 | −0.0673 | −2.1909 | SND | −0.1529 | 1.5850 | −0.0965 | SD | 0.0458 | 0.0854 | 0.5365 | WD |
| 2016–2017 | 35 | 0.1139 | 1.0146 | 0.1123 | WD | 22.2128 | −0.2056 | −108.0512 | SND | 0.5838 | −0.1520 | −3.8405 | SND |
| 2016–2017 | 36 | −0.0428 | −0.0323 | 1.3244 | RD | −0.1190 | 0.4759 | −0.2500 | SD | −0.0812 | 0.0345 | −2.3540 | SD |
| 2016–2017 | 37 | −0.3040 | −0.2323 | 1.3088 | RD | 0.6344 | 9.4036 | 0.0675 | WD | 3.8703 | −0.0397 | −97.4965 | SND |
| 2016–2017 | 38 | −0.0837 | −0.0278 | 3.0084 | RD | 0.0258 | 0.3767 | 0.0684 | WD | −0.1710 | 0.5792 | −0.2952 | SD |
| 2016–2017 | 39 | −0.3489 | 0.2112 | −1.6518 | SD | 1.3365 | 2.3222 | 0.5755 | WD | −0.8791 | −0.3968 | 2.2156 | RD |
| 2016–2017 | 40 | 4.9610 | 0.1538 | 32.2598 | END | −0.9831 | −0.9577 | 1.0265 | RC | 0.0246 | −0.0182 | −1.3525 | SND |
| 2016–2017 | 41 | 0.7211 | 0.8073 | 0.8932 | EC | 0.1648 | 0.1811 | 0.9102 | EC | 0.0289 | 0.0977 | 0.2956 | WD |
| 2016–2017 | 42 | −0.4930 | −0.1270 | 3.8804 | RD | 0.9698 | −0.6934 | −1.3986 | SND | −0.3775 | −0.0653 | 5.7838 | RD |
| 2016–2017 | 43 | 86.8205 | 13.5568 | 6.4042 | END | – | – | – | SD | −0.9929 | −0.8830 | 1.1245 | RC |

**Table A2.** The decoupling statuses of manufacturing subindustries of three regions in 2017–2018.

| Period | Industry | Southern Jiangsu | | | | Northern Jiangsu | | | | Middle Jiangsu | | | |
|---|---|---|---|---|---|---|---|---|---|---|---|---|---|
| | | % C | % GDP | D | State | % C | % GDP | D | State | % C | % GDP | D | State |
| 2017–2018 | 13 | −0.1008 | 0.1502 | −0.6706 | SD | 0.0481 | 0.1057 | 0.4554 | WD | −0.0977 | −0.0266 | 3.6710 | RD |
| 2017–2018 | 14 | 0.4785 | 0.1261 | 3.7939 | END | 0.0044 | 0.2377 | 0.0184 | WD | −0.0858 | 0.0547 | −1.5691 | SD |
| 2017–2018 | 15 | 0.0589 | −0.1517 | −0.3882 | SND | −0.1337 | −0.0861 | 1.5525 | RD | 0.0899 | −0.6864 | −0.1309 | SND |
| 2017–2018 | 16 | −0.0708 | 0.0000 | 0.0000 | SD | 4.2689 | 1.1384 | 3.7501 | END | – | – | – | SD |
| 2017–2018 | 17 | 0.0040 | 0.0511 | 0.0792 | WD | −0.0481 | −0.0924 | 0.5208 | WND | −0.0078 | 0.0222 | −0.3494 | SD |
| 2017–2018 | 18 | 0.1894 | 0.8769 | 0.2160 | WD | 0.7014 | 0.9786 | 0.7168 | WD | 0.2927 | −0.0740 | −3.9545 | SND |
| 2017–2018 | 19 | 0.9807 | −0.0098 | −100.3330 | SND | 0.1253 | 0.0391 | 3.2041 | END | −0.1537 | 0.0577 | −2.6647 | SD |
| 2017–2018 | 20 | 0.0043 | 0.0204 | 0.2118 | WD | 0.1600 | 0.1163 | 1.3752 | END | −0.1870 | −0.0330 | 5.6674 | RD |
| 2017–2018 | 21 | −0.4063 | 0.1478 | −2.7497 | SD | −0.9137 | 0.0222 | −41.1774 | SD | 0.0000 | 0.2010 | 0.0000 | SD |
| 2017–2018 | 22 | −0.0081 | −0.0373 | 0.2160 | WND | −0.2289 | −0.2786 | 0.8215 | RC | −0.4943 | −0.0768 | 6.4382 | RD |
| 2017–2018 | 23 | 0.1870 | 0.1323 | 1.4139 | END | −0.9084 | −0.2393 | 3.7961 | RD | 8.9570 | 26.1612 | 0.3424 | WD |
| 2017–2018 | 24 | 0.3589 | 0.0916 | 3.9178 | END | −0.0280 | 0.1521 | −0.1838 | SD | −0.1676 | −0.0209 | 8.0081 | RD |
| 2017–2018 | 25 | 0.2314 | 0.0977 | 2.3689 | END | 0.2007 | 0.1748 | 1.1478 | EC | −0.0102 | −0.1395 | 0.0729 | WND |
| 2017–2018 | 26 | −0.0944 | −0.3329 | 0.2835 | WND | −0.0617 | 0.2173 | −0.2838 | SD | −0.0826 | 0.0454 | −1.8185 | SD |
| 2017–2018 | 27 | 0.2261 | −0.1980 | −1.1420 | SND | 0.2500 | 0.0700 | 3.5745 | END | −0.1519 | 0.0326 | −4.6639 | SD |
| 2017–2018 | 28 | 0.0685 | 0.0254 | 2.6973 | END | −0.2861 | −0.0409 | 6.9907 | RD | 0.0843 | −0.0405 | −2.0830 | SND |
| 2017–2018 | 29 | 0.0831 | −0.0572 | −1.4521 | SND | −0.0138 | 0.0504 | −0.2732 | SD | −0.0022 | −0.0299 | 0.0737 | WND |
| 2017–2018 | 30 | −0.3209 | −0.1828 | 1.7551 | RD | 0.0941 | −0.0305 | −3.0830 | SND | −0.2030 | 0.1334 | −1.5216 | SD |
| 2017–2018 | 31 | 0.0944 | −0.1044 | −0.9038 | SND | −0.0864 | −0.0803 | 1.0767 | RC | −0.1980 | −0.6358 | 0.3115 | WND |
| 2017–2018 | 32 | 0.0744 | −0.0275 | −2.7021 | SND | −0.0859 | 0.3537 | −0.2427 | SD | 3.3627 | 1.5898 | 2.1152 | END |
| 2017–2018 | 33 | −0.1420 | −0.0433 | 3.2750 | RD | 0.1653 | 0.1950 | 0.8478 | EC | 0.0699 | 0.0583 | 1.1984 | EC |
| 2017–2018 | 34 | −0.1315 | 0.0591 | −2.2258 | SD | −0.1128 | 0.5436 | −0.2075 | SD | −0.0988 | 0.0389 | −2.5369 | SD |
| 2017–2018 | 35 | 0.0813 | 0.2345 | 0.3466 | WD | −0.9877 | −0.4270 | 2.3133 | RD | 0.2763 | −0.0237 | −11.6376 | SND |
| 2017–2018 | 36 | −0.3015 | 0.1262 | −2.3894 | SD | 0.0008 | 0.1638 | 0.0050 | WD | −0.1365 | 2.1902 | −0.0623 | SD |
| 2017–2018 | 37 | 0.1464 | −0.0500 | −2.9275 | SND | −0.1458 | 0.1152 | −1.2655 | SD | −0.9376 | −0.1099 | 8.5334 | RD |
| 2017–2018 | 38 | 3.7538 | 0.1121 | 33.4831 | END | −0.1403 | −0.1455 | 0.9640 | RC | 0.0484 | −0.0499 | −0.9695 | SND |
| 2017–2018 | 39 | −0.1132 | −0.0364 | 3.1106 | RD | 0.1002 | 0.3471 | 0.2887 | WD | −0.1326 | 0.0225 | −5.9039 | SD |
| 2017–2018 | 40 | −0.7024 | −0.2392 | 2.9367 | RD | −0.0603 | −0.8335 | 0.0723 | WND | 0.0071 | 0.3042 | 0.0233 | WD |
| 2017–2018 | 41 | 0.0950 | 0.9577 | 0.0992 | WD | −0.0636 | −0.8531 | 0.0745 | WND | 0.1833 | 0.0455 | 4.0238 | END |
| 2017–2018 | 42 | 0.0839 | 0.0787 | 1.0664 | EC | −0.0029 | 0.3233 | −0.0091 | SD | −0.0092 | −0.0167 | 0.5487 | WND |
| 2017–2018 | 43 | −0.7456 | −0.4766 | 1.5646 | RD | −0.3083 | 1.3331 | −0.2313 | SD | −0.8700 | −0.5858 | 1.4852 | RD |

**Table A3.** The decoupling statuses of manufacturing subindustries of three regions in 2018–2019.

| Period | Industry | Southern Jiangsu | | | | Northern Jiangsu | | | | Middle Jiangsu | | | |
|---|---|---|---|---|---|---|---|---|---|---|---|---|---|
| | | % C | % GDP | D | State | % C | % GDP | D | State | % C | % GDP | D | State |
| 2018–2019 | 13 | −0.0050 | −0.0735 | 0.0676 | WND | 0.2537 | 0.0943 | 2.6891 | END | −0.0293 | −0.0487 | 0.6018 | WND |
| 2018–2019 | 14 | −0.3007 | −0.0071 | 42.5305 | RD | 0.0109 | 0.1040 | 0.1052 | WD | −0.0228 | 0.4005 | −0.0570 | SD |
| 2018–2019 | 15 | 1.1817 | −0.1760 | −6.7159 | SND | −0.0766 | 0.2110 | −0.3632 | SD | 0.1508 | 0.6606 | 0.2283 | WD |
| 2018–2019 | 16 | −0.2017 | 0.0000 | 0.0000 | SD | −0.6424 | 0.6984 | −0.9198 | SD | − | − | − | SD |
| 2018–2019 | 17 | −0.0121 | 0.0324 | −0.3749 | SD | 0.1737 | 0.0530 | 3.2760 | END | −0.0170 | −0.0762 | 0.2235 | WND |
| 2018–2019 | 18 | −0.0802 | −0.0108 | 7.4047 | RD | −0.0884 | −0.2656 | 0.3329 | WND | 0.4570 | −0.2082 | −2.1946 | SND |
| 2018–2019 | 19 | 0.1987 | −0.0192 | −10.3270 | SND | 0.1020 | 0.1115 | 0.9154 | EC | 0.2243 | −0.0660 | −3.3962 | SND |
| 2018–2019 | 20 | 0.4903 | 0.4631 | 1.0587 | EC | 0.0169 | 0.0174 | 0.9665 | EC | −0.0166 | 0.0556 | −0.2992 | SD |
| 2018–2019 | 21 | 0.2463 | 0.0676 | 3.6429 | END | −0.4643 | −0.4195 | 1.1069 | RC | −0.0028 | 0.0000 | 0.0000 | SD |
| 2018–2019 | 22 | −0.0871 | −0.0263 | 3.3120 | RD | 0.0432 | 0.7564 | 0.0571 | WD | 0.8502 | 0.2935 | 2.8972 | END |
| 2018–2019 | 23 | 4.1583 | −0.2928 | −14.2007 | SND | 7.8803 | 0.1749 | 45.0596 | END | 0.2004 | 4.5933 | 0.0436 | WD |
| 2018–2019 | 24 | 0.0633 | 0.0095 | 6.6495 | END | 2.9608 | 1.0243 | 2.8905 | END | 0.0249 | 0.1407 | 0.1771 | WD |
| 2018–2019 | 25 | 0.0133 | 0.0137 | 0.9728 | EC | 0.0157 | −0.0891 | −0.1761 | SND | −0.0458 | −0.1245 | 0.3675 | WND |
| 2018–2019 | 26 | 0.0853 | 0.0470 | 1.8163 | END | 0.0054 | 0.0044 | 1.2215 | END | 0.1235 | 0.0475 | 2.5987 | END |
| 2018–2019 | 27 | 0.3036 | 0.0994 | 3.0546 | END | 0.2122 | −0.1329 | −1.5970 | SND | 0.1007 | 0.2270 | 0.4437 | WD |
| 2018–2019 | 28 | 0.1386 | −0.0273 | −5.0804 | SND | −0.0447 | −0.0512 | 0.8721 | RC | −0.0474 | −0.0671 | 0.7063 | WND |
| 2018–2019 | 29 | 0.1591 | 0.0178 | 8.9364 | END | 0.3080 | 0.1385 | 2.2234 | END | 0.0787 | −0.1209 | −0.6506 | SND |
| 2018–2019 | 30 | −0.1694 | 0.0001 | −1131.7328 | SD | 0.0031 | 0.4563 | 0.0068 | WD | −0.0202 | −0.0137 | 1.4723 | RD |
| 2018–2019 | 31 | −0.0849 | −0.0392 | 2.1671 | RD | 0.3854 | −0.0080 | −48.1417 | SND | 0.1628 | 0.4169 | 0.3905 | WD |
| 2018–2019 | 32 | −0.2145 | −0.2213 | 0.9692 | RC | −0.0132 | 0.2541 | −0.0518 | SD | −0.7926 | −0.5725 | 1.3845 | RD |
| 2018–2019 | 33 | 0.0907 | 0.1513 | 0.5992 | WD | 0.2460 | 0.1948 | 1.2627 | WD | 0.0003 | 0.0030 | 0.0833 | WD |
| 2018–2019 | 34 | 0.2415 | 0.1539 | 1.5694 | END | −0.4937 | 0.1599 | −3.0887 | SD | 3.9916 | 0.5455 | 7.3174 | END |
| 2018–2019 | 35 | 0.3115 | −0.0507 | −6.1377 | SND | 0.1637 | 0.2085 | 0.7851 | WD | 0.9511 | 0.2134 | 4.4560 | END |
| 2018–2019 | 36 | 0.1736 | 0.0123 | 14.1035 | END | −0.0027 | 0.0002 | −16.1790 | SD | 10.2368 | 0.0123 | 830.2161 | END |
| 2018–2019 | 37 | 4.2307 | 0.0584 | 72.4684 | END | 0.7059 | −0.5101 | −1.3839 | SND | 0.0799 | 0.0124 | 6.4478 | END |
| 2018–2019 | 38 | −0.4087 | −0.0728 | 5.6122 | RD | 0.1627 | −0.0663 | −2.4537 | SND | 0.0301 | −0.0164 | −1.8350 | SND |
| 2018–2019 | 39 | −0.0128 | 0.0378 | −0.3392 | SD | −0.2654 | 0.1346 | −1.9713 | SD | 0.1109 | 0.0317 | 3.4996 | END |
| 2018–2019 | 40 | 0.5048 | 0.1749 | 2.8869 | END | 0.9309 | 40.3453 | 0.0231 | WD | 0.0020 | −0.0918 | −0.0215 | SND |
| 2018–2019 | 41 | −0.0035 | 0.1312 | −0.0267 | SD | 0.1826 | 0.5829 | 0.3133 | WD | 0.1400 | −0.2000 | −0.7001 | SND |
| 2018–2019 | 42 | −0.5212 | −0.0330 | 15.8109 | RD | 0.5122 | −0.2901 | −1.7655 | SND | 0.0189 | 0.0910 | 0.2079 | WD |
| 2018–2019 | 43 | 0.3286 | −0.0349 | −9.4224 | SND | 0.1836 | 0.8197 | 0.2240 | WD | 138.8960 | 12.8453 | 10.8129 | END |

**Table A4.** The decoupling statuses of manufacturing subindustries of three regions in 2019–2020.

| Period | Industry | Southern Jiangsu | | | | Northern Jiangsu | | | | Middle Jiangsu | | | |
|---|---|---|---|---|---|---|---|---|---|---|---|---|---|
| | | % C | % GDP | D | State | % C | % GDP | D | State | % C | % GDP | D | State |
| 2019–2020 | 13 | 0.1233 | −0.1278 | −0.9644 | SND | 0.5086 | 0.0807 | 6.2992 | END | 0.2054 | −0.0382 | −5.3778 | SND |
| 2019–2020 | 14 | 0.1503 | 0.0385 | 3.9037 | END | 0.0183 | −0.0224 | −0.8162 | SND | 0.3115 | 1.5709 | 0.1983 | WD |
| 2019–2020 | 15 | −0.0455 | 0.3191 | −0.1425 | SD | −0.0883 | 0.1439 | −0.6139 | SD | 0.5918 | 0.2458 | 2.4077 | END |
| 2019–2020 | 16 | −0.0458 | 0.0000 | 0.0000 | SD | 0.2245 | 0.0116 | 19.3086 | END | − | − | − | SD |
| 2019–2020 | 17 | 0.0021 | 0.0415 | 0.0509 | WD | −0.0911 | 0.1508 | −0.6044 | SD | 0.0478 | 0.0597 | 0.8008 | EC |
| 2019–2020 | 18 | 1.0643 | 0.0060 | 176.3757 | END | 0.5684 | 1.0607 | 0.5359 | WD | −0.3881 | −0.5166 | 0.7514 | WND |
| 2019–2020 | 19 | −0.0470 | 0.0756 | −0.6209 | SD | 0.7085 | 0.6693 | 1.0586 | EC | 7.8090 | 0.1684 | 46.3834 | END |
| 2019–2020 | 20 | 11.9393 | −0.1596 | −74.8185 | SND | 0.3452 | 0.2949 | 1.1705 | EC | −0.4579 | −0.0399 | 11.4857 | RD |
| 2019–2020 | 21 | 0.0809 | 0.0374 | 2.1611 | END | 0.0000 | 0.0000 | 0.0000 | SD | 0.3588 | 0.3347 | 1.0722 | EC |
| 2019–2020 | 22 | 0.1555 | −0.0136 | −11.4419 | SND | −0.1782 | −0.0446 | 3.9928 | RD | 0.3948 | 0.3834 | 1.0297 | EC |
| 2019–2020 | 23 | −0.8491 | 0.2735 | −3.1048 | SD | 0.0428 | 0.0992 | 0.4313 | WD | −0.5818 | −0.9191 | 0.6330 | WND |
| 2019–2020 | 24 | 0.1117 | 0.0311 | 3.5951 | END | 0.0000 | 0.0036 | 0.0000 | SD | 0.0333 | 0.3914 | 0.0852 | WD |
| 2019–2020 | 25 | −0.0114 | −0.1521 | 0.0751 | WND | 0.4550 | 0.1007 | 4.5170 | END | −0.0258 | −0.4682 | 0.0551 | WND |
| 2019–2020 | 26 | 0.0692 | −0.0143 | −4.8292 | SND | 0.7221 | 0.1515 | 4.7666 | END | 0.0265 | −0.0713 | −0.3720 | SND |
| 2019–2020 | 27 | 2.5993 | 0.0968 | 26.8540 | END | −0.3489 | 0.9757 | −0.3576 | SD | 0.4795 | 2.5025 | 0.1916 | WD |
| 2019–2020 | 28 | 0.0908 | 0.1961 | 0.4633 | WD | 0.9813 | −0.1078 | −9.1072 | SND | −0.0292 | −0.0536 | 0.5450 | WND |
| 2019–2020 | 29 | 0.0303 | 0.1559 | 0.1944 | WD | −0.0702 | −0.0386 | 1.8189 | RD | −0.0556 | 1.8732 | −0.0297 | SD |
| 2019–2020 | 30 | −0.0258 | 0.2100 | −0.1226 | SD | −0.0931 | −0.0808 | 1.1526 | RC | 0.0303 | 0.2870 | 0.1057 | WD |
| 2019–2020 | 31 | 0.1018 | −0.1522 | −0.6688 | SND | 0.5420 | 0.1698 | 3.1923 | END | 3.1528 | 0.1442 | 21.8580 | END |
| 2019–2020 | 32 | 0.0475 | −0.0199 | −2.3887 | SND | 0.1695 | −0.3351 | −0.5059 | SND | 0.0676 | 0.0835 | 0.8090 | EC |
| 2019–2020 | 33 | −0.1307 | −0.1445 | 0.9045 | RC | −0.1238 | −0.0569 | 2.1739 | RD | 0.2330 | 0.0857 | 2.7206 | END |
| 2019–2020 | 34 | −0.2631 | −0.1460 | 1.8025 | RD | 0.1137 | −0.2919 | −0.3894 | SND | 0.0417 | −0.1453 | −0.2869 | SND |
| 2019–2020 | 35 | −0.1828 | 0.1951 | −0.9368 | SD | −0.5284 | −0.1963 | 2.6913 | RD | −0.2245 | 0.4129 | −0.5437 | SD |
| 2019–2020 | 36 | 0.0100 | 0.1143 | 0.0875 | WD | 0.5022 | 0.9216 | 0.5449 | WD | −0.8739 | −0.5939 | 1.4716 | RD |
| 2019–2020 | 37 | −0.7033 | −0.0578 | 12.1687 | RD | −0.6702 | 0.0876 | −7.6542 | SD | 1.0600 | 0.0770 | 13.7650 | END |
| 2019–2020 | 38 | −0.4789 | 0.0766 | −6.2513 | SD | −0.0355 | 0.1212 | −0.2932 | SD | −0.2649 | 0.2124 | −1.2468 | SD |
| 2019–2020 | 39 | 0.0786 | −0.0683 | −1.1510 | SND | 0.3994 | 0.5984 | 0.6674 | WD | 0.6285 | 0.4692 | 1.3395 | END |
| 2019–2020 | 40 | 0.0853 | 0.1011 | 0.8433 | EC | 4.0370 | −0.3195 | −12.6362 | SND | 0.1151 | 0.3449 | 0.3337 | WD |
| 2019–2020 | 41 | 5.2892 | −0.2930 | −18.0539 | SND | 6.0398 | 0.5641 | 10.7068 | END | 1.5307 | 0.7020 | 2.1805 | END |
| 2019–2020 | 42 | −0.5716 | −0.0953 | 5.9966 | RD | −0.2749 | 1.9982 | −0.1376 | SD | 5.0293 | 0.1072 | 46.9108 | END |
| 2019–2020 | 43 | 0.5929 | 3.5638 | 0.1664 | WD | 0.7335 | 0.4119 | 1.7807 | END | 15.1724 | −0.4976 | −30.4908 | SND |

**Table A5.** Industrial classification for the manufacturing industry.

| Code | Name |
|------|------|
| 13 | Farm and sideline food procarbon emissions |
| 14 | Food |
| 15 | Beverage |
| 16 | Tobacco |
| 17 | Textiles |
| 18 | Textiles and garments, shoes, hats |
| 19 | Leather, fur, feathers and other products |
| 20 | Wood procarbon emissions and furniture making |
| 21 | Furniture |
| 22 | Paper making and paper products |
| 23 | Copies of printing and recording mediums |
| 24 | Cultural and educational sporting goods |
| 25 | Oil procarbon emissions, coking and nuclear fuel |
| 26 | Chemical raw materials and chemical products |
| 27 | Pharmaceuticals |
| 28 | Chemical fiber |
| 29 | Rubber products |
| 30 | Plastic products |
| 31 | Nonmetallic mineral products |
| 32 | Ferrous metal smelting and rolling |
| 33 | Nonferrous metal smelting and rolling |
| 34 | Fabricated metal products |
| 35 | General machinery |
| 36 | Equipment for special purposes |
| 37 | Transportation equipment |
| 38 | Electrical equipment and machinery |
| 39 | Communication equipment, computer and other |
| 40 | Instrumentation, stationary and office supplies |
| 41 | Other manufacturing |
| 42 | Waste-resource carbon-emission comprehensive-utilization industry |
| 43 | Metal products, machinery and equipment repair |

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
