# Peer review of "Carbon Emissions from Manufacturing Sector in Jiangsu Province: Regional Differences and Decomposition of Driving Factors"

_sustainability, doi:10.3390/su14159123_

Round 1

Reviewer 1 Report

Review

Comments to the Authors:

This manuscript presents studies on calculations of the decoupling statuses between manufacturing industry CEs and IAV in southern Jiangsu, northern Jiangsu and middle Jiangsu from 2016 to 2020. The research achievements are interesting and promising. On the other hand, after a careful and detailed analysis there are some suggestions and questions to the authors. Thus, it is recommended to read all manuscript carefully and make corrections.

1)       Page 1, lines 33 – 35: In my opinion, current literature data should be presented until 2022 on the global emissions of carbon dioxide into the atmosphere and the leading countries in this regard. What are the long-term goals for reducing CO2 emissions in the world and in China?

2)       Pages 1 - 2, lines 31 – 46: References should be added at the end of this paragraph.

3)       Page 2, lines 47 – 63: References should be added at the end of this paragraph.

4)       Page 2, lines 64 – 75: References should be added in the paragraph.

5)       Pages 6 – 12, lines 200 - 427: Is it possible to carry out a statistical analysis of the results? If so, it is recommended to perform this analysis.

6)       Page 7, line 235: There is a grammatical error in the sentence: ‘Table 3 show the decomposition (…)’. It should be corrected.

7)       Page 10, line 373: In my opinion, the phrase ‘Compared with’ should be replaced by ‘Compared to’.

8)       Page 11, line 386: In my opinion, the phrase ‘First’ should be replaced by ‘Firstly’.

9)       Page 11, line 387: There is a grammatical error in the sentence: ‘Table 5 show the analysis results.’. It should be corrected.

10)   Page 11, line 399: In my opinion, the phrase ‘Second’ should be replaced by ‘Secondly’.

11)   Page 11, lines 407 - 408: There is a grammatical error in the sentence: ‘Table 7 show the results.’. It should be corrected.

12)   Pages 12 - 14, lines 427 – 507: Were there similar analyzes performed in other regions of China or other countries in the world? It is recommended to carry out a literature review and if there are similar analyzes a comparison should be made and cited.

13)   Pages 1 – 15, lines 8 - 557: It is recommended to check the entire text for correctness of English language by an English philology specialist or a native speaker.

14)   Pages 15 – 17, lines 558 – 699: The references should be improved and some data is missing in several ones. Please, correct them, complete the information and rewrite them in the form in accordance with the editorial requirements of the Sustainability journal.

Author Response

Dear reviewer,

Thank you very much for your affirmation of our work. Also, we would like to thanks for spending time on reading our manuscript and providing these constructive suggestions to help us improve the quality of our manuscript. We have tried to address those issues. And the point-by-point responses in attachment. Thank you again for this valuable comment. We hope that you will be satisfied with our response.

Reviewer 2 Report

The article addresses important and current problems related to low-carbon transformation on the example of decomposing the decoupling of carbon emissions and industrial growth of manufacturing in Jiangsu Province. However, before accepting it, there will be a need to make some adjustments.

Please adapt the manuscript to the journal's requirements, eg References.

Please consider if some of the sentences in Conclusions and policy implications are not too obvious on the site? E.g. "Promote strategic emerging industries’ development within the region." and "Constantly promoting the development of strategic emerging industries within a region is conducive to the high-quality development of regional manufacturing and the reduction in CEs.".

Author Response

Dear reviewer,

Thank you very much for your affirmation of our work. Also, we would like to thanks for spending time on reading our manuscript and providing these constructive suggestions to help us improve the quality of our manuscript. We have tried to address those issues. And  the point-by-point responses in attachment. Thank you again for this valuable comment. We hope that you will be satisfied with our response.

Best wishes for you!

Reviewer 3 Report

Dear authors,

Thank you for preparing an interesting manuscript on decarbonisation conditions in Jiangsu Province.

The abstract should be improved. The current abstract is too vague and only provides little information. For example, this first sentence tells nothing. You need to introduce the background, research gaps, your research objectives, research methods, your key findings (as quantitative research, there is not even a number in your abstract) and suggestions for future research in the abstract. Please rewrite the abstract and include all the items mentioned.

Introduction can be improved. You need one key/guiding research question and a clear objective. Do you want to assess the decarbonisation routes of manufacturing? Or do you want to understand the challenges embedded? You need a theme! You asked too many research questions but didn’t answer them clearly in the conclusions!

It would be great if you could provide a map of Jiangsu and the three sub-regions of Jiangsu.

In terms of Methodology, you need to explain why the LMDI model could be applied to Jiangsu, as well as its merits and limitations

We study the past to under the future. In the discussion section, you need to include the latest progress and policies of the 14th five-year plan to examine the validity of your analysis.

Policy implications are a bit too vague. Please try to specify and add practical insights. Try to connect latest iconic events and policies.

Last but not least, there are too many acronyms! That makes your paper very hard to read and follow. Cut unnecessary ones or replace them with general words. For example, use emissions to replace carbon emissions, rather than CEs.

Some minor comments are listed below.

Lines 8: carbon neutrality is in 2060, not 2030

Lines 40-42: reference is needed for the key statistics

Line 41: full spelling GDP

Line 42: to assertive to say ‘results are not satisfactory’. Why? Please provide some data and explain

Line 47-52: references needed

Lines 53: Jiangsu seems to have finished most industrialisation and urbanisation, what do you mean by ‘a critical period’ downstream or upstream?

Line 87: coupling NOT decoupling. decoupling refers to an economy that would be able to grow without corresponding increases in environmental pressure

Line 142: delete ‘existing’

Table 6: Capitalise all first letters in the first column

Author Response

(The authors gave the same response as above.)

Round 2

Reviewer 1 Report

Thank you for improving the manuscript according to the Reviewers' suggestions.

Author Response

Dear reviewer,

Thank you very much for your affirmation of our work. Also, we would like to thanks for spending time on reading our manuscript and providing these constructive suggestions to help us improve the quality of our manuscript. We read through the whole manuscript many times and corrected spelling, grammar and other mistakes and reorganized the language to make it more fluid in this paper. Thank you again for this important comment. We hope that you will be satisfied with our response.

Best wishes for you

Hailing 

Reviewer 3 Report

Dear authors,

Thank you very much for your careful preparation of the revised manuscript and for addressing our feedback. There is a bit more work that needs to be done. Please find my suggestions below. 

Line 4: is the second author's name correct? 

line 30: cite academic references about 30-60 targets, such as https://doi.org/10.1088/1748-9326/ac30bf or 

doi.org/10.1016/j.enpol.2021.112350 

line 30: Change 'in order to' to 'to'

line 66; emissions 

line 82 change 'have been paying attention to' to 'pay attention to'

Start from 996, change all beginning verbs to nouns. For example, use clarifying Not clarify, use being combined with NOT combined with. 

Author Response

Dear reviewer,

Thank you very much for your affirmation of our work. Also, we would like to thanks for spending time on reading our manuscript and providing these constructive suggestions to help us improve the quality of our manuscript. We have tried to address those issues. And below are the point-by-point responses. Thank you again for this important comment. We hope that you will be satisfied with our response.

Point 1: cite academic references about 30-60 targets, such as doi.org/10.1088/1748-9326/ac30bf or doi.org/10.1016/j.enpol.2021.112350

Response 1: Thanks for your precious advice. We read the paper "The dark side of ambition: side-effects of China’s climate policy." and "China's climate ambition: Revisiting its First Nationally Determined Contribution and centering a just transition to clean energy" according to your recommendation. We found that these two papers are very detailed on 30-60 targets, which has important reference significance for the study of this paper. 

Thank you again for this valuable comment. We hope that you will be satisfied with our response.

Point 2:

line 30: Change 'in order to' to 'to'

line 66; emissions

line 82 change 'have been paying attention to' to 'pay attention to'

Start from 996, change all beginning verbs to nouns. For example, use clarifying Not clarify, use being combined with NOT combined with.

Response 2: Thanks for your helpful comment. According to your suggestion, we read through the whole manuscript many times and corrected spelling, grammar and other mistakes. And we reorganized the language to make it more fluid in this paper . 

Thank you again for this valuable comment. We hope that you will be satisfied with our response.

Best wishes for you